# Statistical Road-Traffic Noise Mapping Based on Elementary Urban Forms in Two Cities of South Korea

**Phillip Kim** [1], **Hunjae Ryu** [2], **Jong-June Jeon** [3] **and Seo Il Chang** [4,*]

1 Department of Energy and Environmental System Engineering, University of Seoul, Seoul 02504, Korea; rlavlfflq154@uos.ac.kr
2 Korean Educational Environments Protection Agency, Chungcheongbuk-do 28166, Korea; pgryuno1@schoolkeepa.or.kr
3 Department of Statistics & Graduate School, Department of Urban Big Data Convergence, University of Seoul, Seoul 02504, Korea; jj.jeon@uos.ac.kr
4 School of Environmental Engineering & Graduate School, Department of Urban Big Data Convergence, University of Seoul, Seoul 02504, Korea
* Correspondence: schang@uos.ac.kr; Tel.: +82-2-6490-2865

**Abstract:** Statistical models that can generate a road-traffic noise map for a city or area where only elementary urban design factors are determined, and where no concrete urban morphology, including buildings and roads, is given, can provide basic but essential information for developing a quiet and sustainable city. Long-term cost-effective measures for a quiet urban area can be considered at early city planning stages by using the statistical road-traffic noise map. An artificial neural network (ANN) and an ordinary least squares (OLS) model were developed by utilizing data on urban form indicators, based on a 3D urban model and road-traffic noise levels from a normal noise map of city A (Gwangju). The developed ANN and OLS models were applied to city B (Cheongju), and the resultant statistical noise map of city B was compared to an existing normal road-traffic noise map of city B. The urban form indicators that showed multi-collinearity were excluded by the OLS model, and among the remaining urban forms, road-related urban form indicators such as traffic volume and road area density were found to be important variables to predict the road-traffic noise level and to design a quiet city. Comparisons of the statistical ANN and OLS noise maps with the normal noise map showed that the OLS model tends to under-estimate road-traffic noise levels, and the ANN model tends to over-estimate them.

**Keywords:** road-traffic noise; urban forms; artificial neural network; statistical noise map





## 1. Introduction

Road-, railway-, and air-traffic networks are essential for the sustainable development of a city [1,2], but transport networks can also generate undesirable by-products including fine dust and noise [3], affecting the quality of life and health of inhabitants [4,5]. The European Union (EU) recommends generating strategic noise maps for efficient management of traffic noise every five years, based on the END (European Noise Directive) 2002/49/EC [6]. The targets of noise maps include major roads [7], railways [8], airports [9,10], and ports [11]. Similar to the EU, the Republic of Korea (South Korea) mandates noise mapping in large cities, and updates every five years [12]. The noise maps are used to establish action plans to reduce the noise exposure of inhabitants in cities [13].

In South Korea, the road-traffic network is the main transportation network [14], and inevitably, high population density causes a lot of residents living near the road-traffic infrastructure to be exposed to high levels of road-traffic noise. Therefore, road-traffic noise is one of the most significant adverse health impacts experienced by the people of South Korea [15]. Various methods are used to reduce road-traffic noise, including porous asphalt, reduced speed limits, and heavy vehicle detours. As a measure for the propagation path,

noise barrier walls and tunnels are used at a relatively high cost. The shapes, material, and specifications of walls and tunnels determine the noise reduction effect. By utilizing a noise map, if it exists, it is possible to find a combination of noise reduction methods and specifications that lead to the best results.

Although the noise map is an efficient tool, if it does not exist already, its generation over a wide area requires abundant GIS (geographic information system) data on roads, buildings, traffic, and so forth, as well as considerable hardware, software, and financial resources. At early urban planning stages, when there is only minimal information on the population, road density, and ground space index, a normal noise map cannot be generated because no concrete data for buildings, roads, and traffic exist.

Therefore, when there is no normal noise map available, a statistical relationship between the elementary urban forms and road-traffic noise can provide useful information for city planning, and a noise map with a low resolution can be generated from the statistical model. The relationship between the road-traffic noise level and urban forms has been studied using various statistical methods, such as land-use regression (LUR) models [16–22], regression models [23,24], and the spatial statistical model [25]. In these cases, the road-traffic noise levels for the statistical analyses were measured [16–22,26] or simulated [23–25,27–30].

Aguilera et al. [16] applied LUR modeling to assess the long-term intraurban spatial variability of road-traffic noise in three European cities. The authors used short-term noise measurements for the three cities to develop two LUR models: one GIS-only and one Best. The GIS-only model considered only predictor variables derived from GIS data, and the Best model considered GIS data and variables collected while visiting the measurement sites. The authors' findings indicate that LUR modeling, with accurate GIS source data, can be a promising tool for noise exposure assessment. Han et al. [17] analyzed the relationships between environmental noise and urban morphology. Socioeconomic data and landscape data were selected as representative of the urban morphology. The environmental noise, including regional environmental noise and traffic, was significantly influenced by urban morphology parameters. Xie et al. [18] applied the LUR model to depict urban environmental noise. Six land-use types and the length of three road types were used as independent variables. The model was fitted with linear and nonlinear structures, but there was no significant difference. Various characteristics of the area, such as traffic, weather, and land use, were used in the LUR modeling to predict the noise level [19–22]. These studies developed the LUR models based on the noise level measured at a specific point.

Salomons and Berghauser Pont [23] presented relationships between the spatial distribution of traffic noise in a city and traffic volume, urban density, and form. The researchers investigated this by means of numeric calculations for two cities and various idealized urban fabrics, and in the two cities, the average sound level in an urban area decreased with increasing population and building density. Silva et al. [24] addressed the problems of the urban environment in the form of interactions between urban forms and urban road-traffic noise monitored using urban form indicators and models for the effects of noise propagation forms. The authors used the compactness index (CI), porosity index (ROS), and complexity perimeter index (fractal) of urban areas as urban form indices and performed a correlation analysis between the average facade noise levels and each index. The correlations were positive between urban road-traffic noise and CI and ROS but negative between urban road-traffic noise and fractals.

Ryu et al. [25] developed spatial statistical models to predict road-traffic noise in cities. Specifically, the authors used a spatial autoregressive model (SAR) and a spatial error model (SEM), in addition to a classical linear model, to account for spatial dependence in determining the road-traffic noise levels in relation to urban form indicators. The SAR model had better statistical properties, although it had slightly less explanatory power than that of the SEM model. The results showed that ground space index, floor space index, traffic volume, speed, road area density, and fraction of industrial area had statistically

significant direct and indirect impacts on road-traffic noise levels. In addition, various statistical analyses between noise and characteristics of the urban area, such as urban forms and morphologies, have been performed in previous studies [26–30].

Artificial neural network (ANN) models have been used to predict road-traffic noise at specific points with traffic flow characteristics (e.g., traffic volume, speed, and heavy vehicle ratio). Kumar et al. [31] fitted an ANN model with measured road-traffic noise levels, traffic volume, speed, and heavy vehicle ratios, and the predicted noise levels using the ANN model were closer to the measured noise levels than were the levels predicted with the ordinary least squares (OLS) model. Garg et al. [32] predicted road-traffic noise levels (Leq, L10) with a total of eight explanatory variables: the volumes of cars, two-wheelers, medium commercial vehicles, three-wheelers, buses, and trucks and the speeds of heavy and light vehicles. The predicted noise levels using the ANN model were closer to the measured noise levels than the predictions from the multiple linear regression analysis. Hamad et al. [33] fitted an ANN model predicting road-traffic noise levels in a city with a hot climate. The ground temperature and acoustical factors, such as distance from noise source (road), traffic volume, and speed, were used. The database was applied to a basic statistical traffic noise model, the Ontario road noise analysis method for the environment and transportation (ORNAMENT), and the ANN model. The predicted road-traffic noise level from the ANN model was closer than the other models to the measured noise level. Similar to these studies, ANN models that predict the road-traffic noise level based on measured traffic data were developed to replace traditional road-traffic noise prediction [34–36].

In this study, ANN and OLS (ordinary least squares) models were developed simultaneously to reflect the relationship between urban form indicators and road-traffic noise levels. The road-traffic noise levels were obtained from a simulated normal noise map of the city of Gwangju, and the urban form indicators were estimated from the 3D city model and relevant GIS data [37]. The developed ANN and OLS models were applied to another city, Cheongju, to generate ANN and OLS statistical noise maps, respectively. The noise levels from the two statistical noise maps were compared with those from a normal noise map of the city of Cheongju.

## 2. Materials and Methods

The study area was divided by applying a grid cell system, and the representative values of urban form indicators and road-traffic noise levels were calculated for each grid cell. The representative values of urban form indicators of a grid cell were extracted based on 3D GIS data, and the representative values of the road-traffic noise level were extracted from an existing normal noise map and verified by on-site measurements [37].

### 2.1. Site Selection and Data Preparation

Gwangju Metropolitan City in South Korea, which was previously noise-mapped, was selected as the study area, and the City of Cheongju was selected as the verification area to compare its predicted road-traffic noise levels. The noise map of Gwangju was produced with noise mapping software, SoundPLAN [38], and includes road, railway, and air traffic as noise sources [39]. Figure 1 shows the road-traffic normal noise map of Gwangju. The total area of Gwangju was 501 km$^2$ as of 2017, when the noise map was developed, and the population was 1.46 million. The total area of Cheongju was 153 km$^2$ as of 2013, when the noise map was developed, and the population was about 0.689 million.

Representative values of the road-traffic noise levels and the urban form indicators were calculated for each grid cell. As the size of a grid cell affects the statistical results, it should be determined by a careful test of different sizes of grid cells. Salomons and Berghauser Pont [23] used 250 m × 250 m grid cells to extract representative values of facade noise levels, traffic volume, and urban density. Chun and Guldmann [40] used three grid cell systems of the sizes of 120 m × 120 m, 240 m × 240 m, and 480 m × 480 m to analyze the impacts of urban characteristics on urban heat islands. The best model was obtained statistically from the smallest cells of 120 m × 120 m. Ryu [41] analyzed

the relationship between road-traffic noise and urban form indicators by varying the size of grid cell systems of 125 m × 125 m, 250 m × 250 m, and 300 m × 300 m and found that the grid cell system of 125 m × 125 m best reflected the characteristics of the urban form indicators.

Since the study and verification areas for this study were Gwangju and Cheongju of South Korea, the 125 m × 125 m grid cell system that showed the best result in Ryu [41] was adopted, and representative values of road-traffic noise and urban form indicators were calculated in each grid cell.

In this study, various types of data were used to generate a 3D urban model and road-traffic noise map. Topography data were used to develop a digital elevation model for a 3D urban model, and road and building data were also used to create the 3D urban model. LiDAR (light detection and ranging) data were used to supplement the building data without height and to model individual overpass roads [37]. LiDAR data were used to increase the precision of the 3D urban model [37,42,43], and through this, accurate production of noise maps and clear reflection of urban characteristics were achieved. Land-use and population data were used to calculate the urban form indicators. A summary of the input data for Gwangju and Cheongju used in this research is shown in Appendix A.

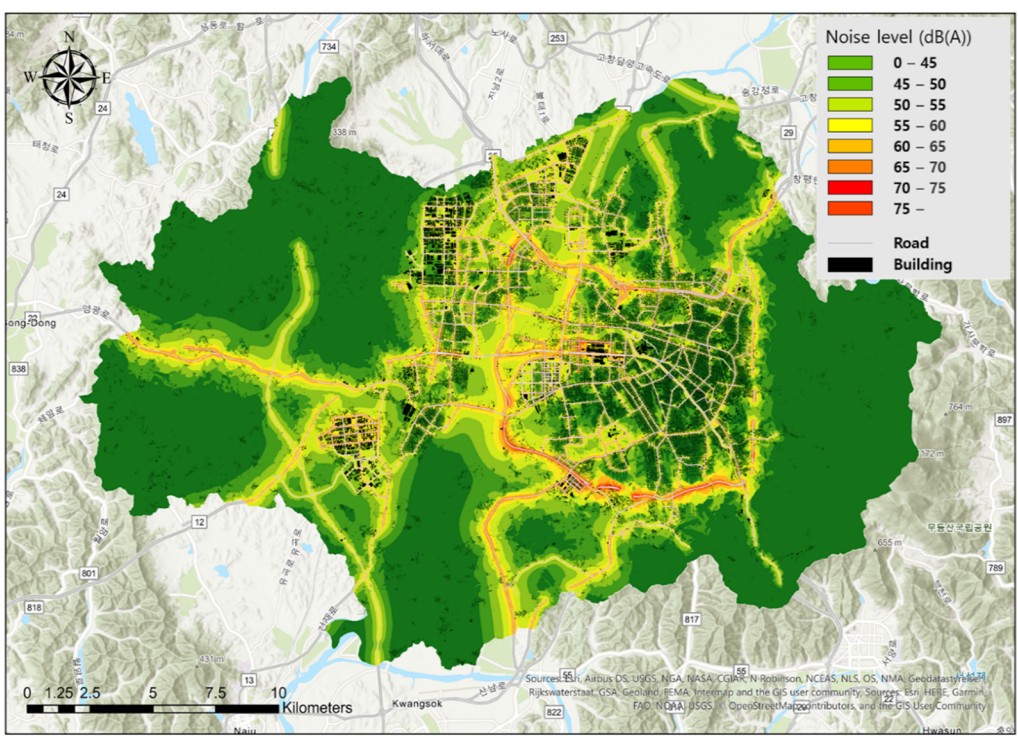

**Figure 1.** Road-traffic noise map of Gwangju Metropolitan City [39,44,45].

*2.2. Representative Values of the Road-Traffic Noise Level and Urban Form Indicators of a Grid Cell*

Representative values of the road-traffic noise level were calculated for each grid cell from an existing noise map [39] by averaging daytime noise levels on every facade of the buildings that were included in the grid cell. In previous studies [23,25], population (inhabitant)–weighted averaging was used to estimate representative values of the road-traffic noise level for grid cells. Calculation of population (inhabitant)–weighted averaging noise levels is suitable for estimating the noise exposure in residential areas. However, the purpose of estimating the representative value of the road-traffic noise level in this study was to understand the noise distribution throughout the city regardless of land use. Therefore, the representative value of the road-traffic noise level for each grid cell should reflect the averaged noise level of all buildings in the grid cell. As shown in Figure 2a, the

grid cell which does not contain any building was not considered for the statistical analysis, and the numbers of valid grid cells were 11,990 for Gwangju and 4173 for Cheongju, respectively. In grid cells that contained only whole buildings, such as in Figure 2b, all the facade noise levels were used to calculate the representative value. For buildings that were straddled over multiple grid cells, as in Figure 2c,d, only the facade noise levels contained in each grid cell were used for the calculation. The representative value of the road-traffic noise level, defined as *L*, was measured as follows:

$$L \equiv 10 \times \log\left\{ \frac{1}{\sum_k \sum_j \sum_i} \left( \sum_k \sum_j \sum_i 10^{\frac{L_{ijk}}{10}} \right) \right\}, \tag{1}$$

where $L_{ijk}$ is the noise level at the facade *i*, floor *j*, and building *k*.

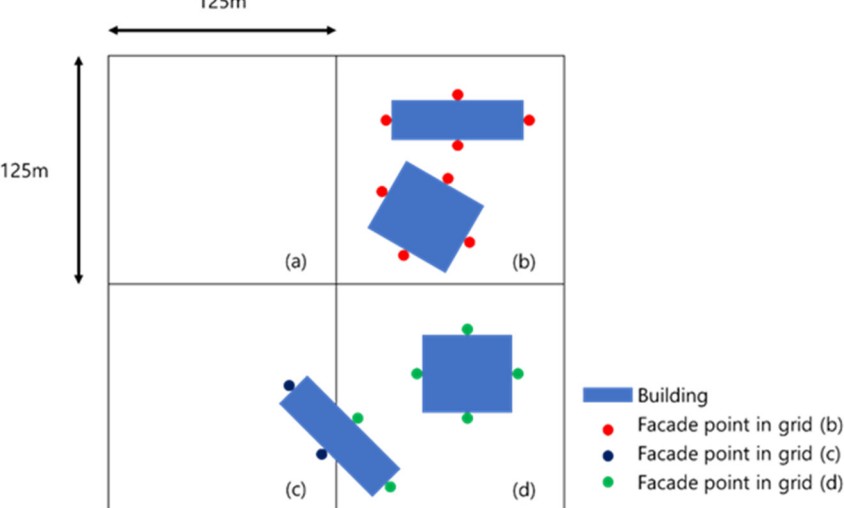

**Figure 2.** Estimation of the representative value of the road-traffic noise level in each grid cell: (**a**) no building in a cell; (**b**) whole buildings in a cell; (**c**) a straddled building over two cells; (**d**) a building straddled over two cells and a whole building in a cell.

The representative values of urban form indicators were calculated with a 3D urban model that was used to create a normal noise map, and the indicators were selected that could affect road-traffic noise emissions and propagation [27,28,41]. The indicators were categorized as population-, building-, road-, and land-use-related. Population density, *P*, was chosen as a population-related urban form indicator. In previous studies [17,23], the population density showed a weak positive or negative correlation with the road-traffic noise level. Population density is defined as follows for a grid cell:

$$P \equiv \frac{\sum_k \sum_j \sum_i P_{ijk}}{A}, \tag{2}$$

where $P_{ijk}$ is the number of persons at the facade *i*, floor *j*, and building *k*, and *A* is the area of the cell.

Ground space index (*GSI*) and floor space index (*FSI*) were selected as building-related urban form indicators. *GSI* is the ratio of the floor area of the buildings in a cell to the total area of the grid cell, and *FSI* is the ratio of the total area of the buildings in a cell to the total area of the grid cell. Buildings in an urban area play a role, with respect to acoustics, in reducing road-traffic noise due to reflection, diffraction, and absorption. In the noise mapping software, input factors such as reflection number and absorption loss of buildings

are used to consider the effect of noise reduction due to the facade of buildings [38,46]. *GSI* and *FSI* are calculated as follows:

$$GSI \equiv \frac{\sum_k G_k}{A},$$

(3)

where $G_k$ is the area of building *k*, and *A* is the cell area, and:

$$FSI \equiv \frac{\sum_K G_k J_k}{A},$$

(4)

where $J_k$ is the number of floors of building *k*.

Road-related urban form indicators were estimated by road-segment-area-weighted averaging. Traffic volume (*Q*), heavy vehicle ratio (*PH*), and speed (*V*) were selected. Traffic volume, heavy vehicle ratio, and speed are core factors that could explain the mechanism of road-traffic noise emissions. The variables are the basic input factors in the mathematical road-traffic noise prediction models, such as RLS-90, NMPB 2008, and TNM [47–49], and are also inputs used in noise mapping software to calculate the emission level of road-traffic noise [38,46]. Traffic volume, heavy vehicle ratio, and speed are defined as follows, respectively:

$$Q \equiv \frac{\sum_i R_i w_i Q_i}{\sum_i R_i w_i},$$

(5)

$$PH \equiv \frac{\sum_i R_i w_i PH_i}{\sum_i R_i w_i},$$

(6)

$$V \equiv \frac{\sum_i R_i w_i V_i}{\sum_i R_i w_i},$$

(7)

where $R_i$, $w_i$, $Q_i$, $PH_i$, and $V_i$ are the length, width, traffic volume, heavy vehicle ratio, and traffic speed of road segment *i*, respectively. Traffic density is defined as:

$$D_t \equiv \frac{Q}{V}$$

(8)

The road area density, $R_a$, and noise barrier area density, $W_a$, were also selected. Road area density is the ratio of the road area in a cell to the total area of the grid cell, and noise barrier area density is the ratio of the noise barrier area in a cell to the total area of the grid cell. The increased road area density tends to increase road-traffic noise emission levels, and the increased noise barrier area density tends to reduce the road-traffic noise level. The road area density and noise barrier density were defined, respectively, as:

$$R_a \equiv \frac{\sum_i R_i w_i}{A},$$

(9)

$$W_a \equiv \frac{\sum_i W_i}{A},$$

(10)

where $R_i$, $w_i$, and $W_i$ are the length and width of road segment *i* and the area of noise barrier *i*, respectively.

Finally, land-use-related urban form indicators were calculated as the fraction of land use in each grid cell. Various studies have been performed to analyze the relationship between land-use characteristics and road-traffic or environmental noise [16–22]. Categories of land use in this study were residential, commercial, industrial, and green area, and were defined, respectively, as:

$$L_R \equiv \frac{LR}{A}, \quad L_C \equiv \frac{LC}{A}, \quad L_I \equiv \frac{LI}{A}, \quad L_G \equiv \frac{LG}{A},$$

(11)

where *LR*, *LC*, *LI*, *LG*, and *A* are the areas of residential, commercial, industrial, green use, and a cell, respectively. The land-use urban form indicators were assumed to satisfy the following condition:

$$1 = L_R + L_C + L_I + L_G . \tag{12}$$

Figures 3 and 4 show the grid maps of representative values for *L*, *P*, *FSI*, and *V* in Gwangju and Cheongju. The grid maps of representative values for other urban form indicators are presented in supplement A (Supplementary Materials).

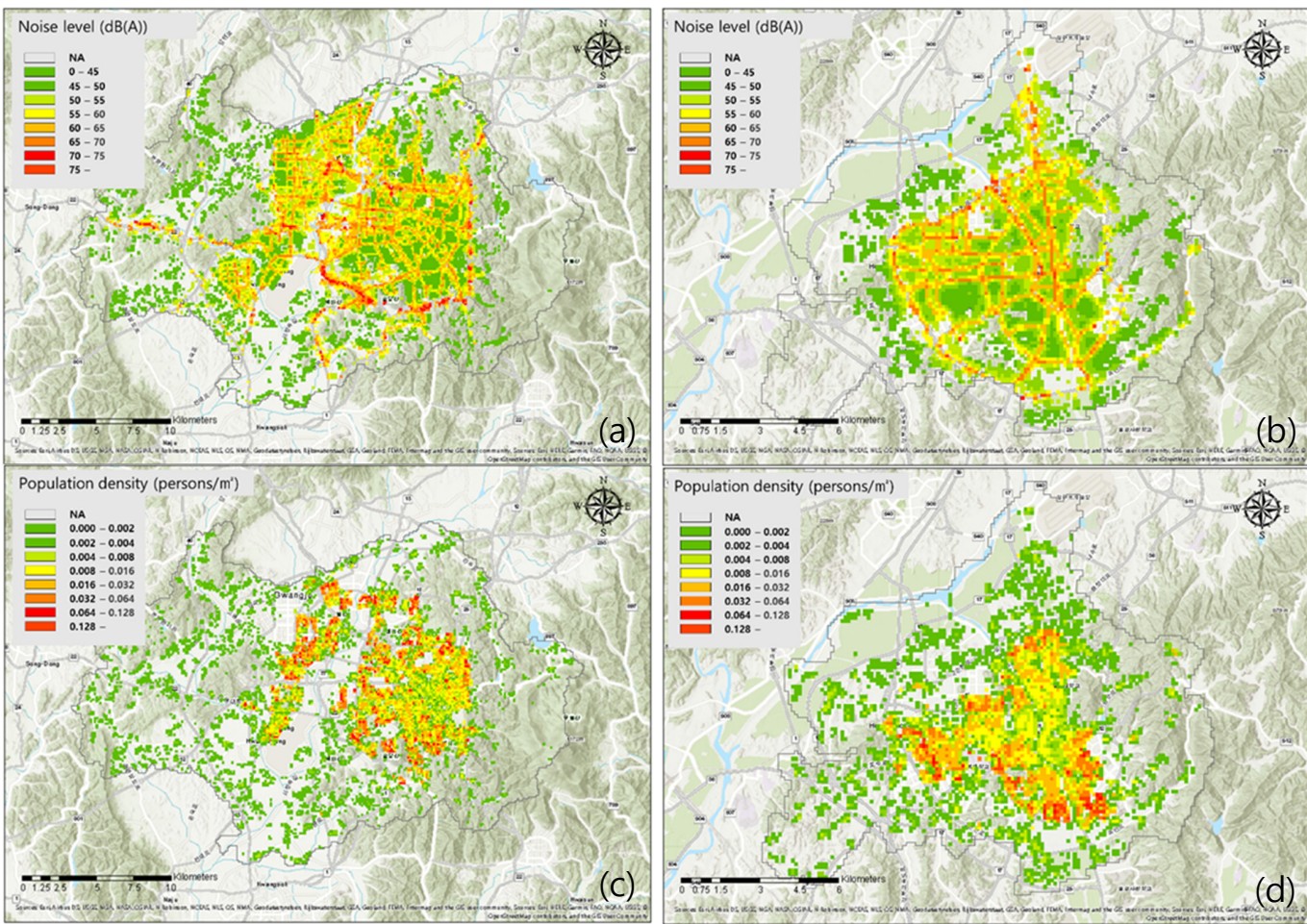

**Figure 3.** Grid maps of: (**a**) *L* (road-traffic noise level) in Gwangju; (**b**) *L* (road-traffic noise level) in Cheongju; (**c**) *P* (population density) in Gwangju; (**d**) *P* (population density) in Cheongju [44,45].

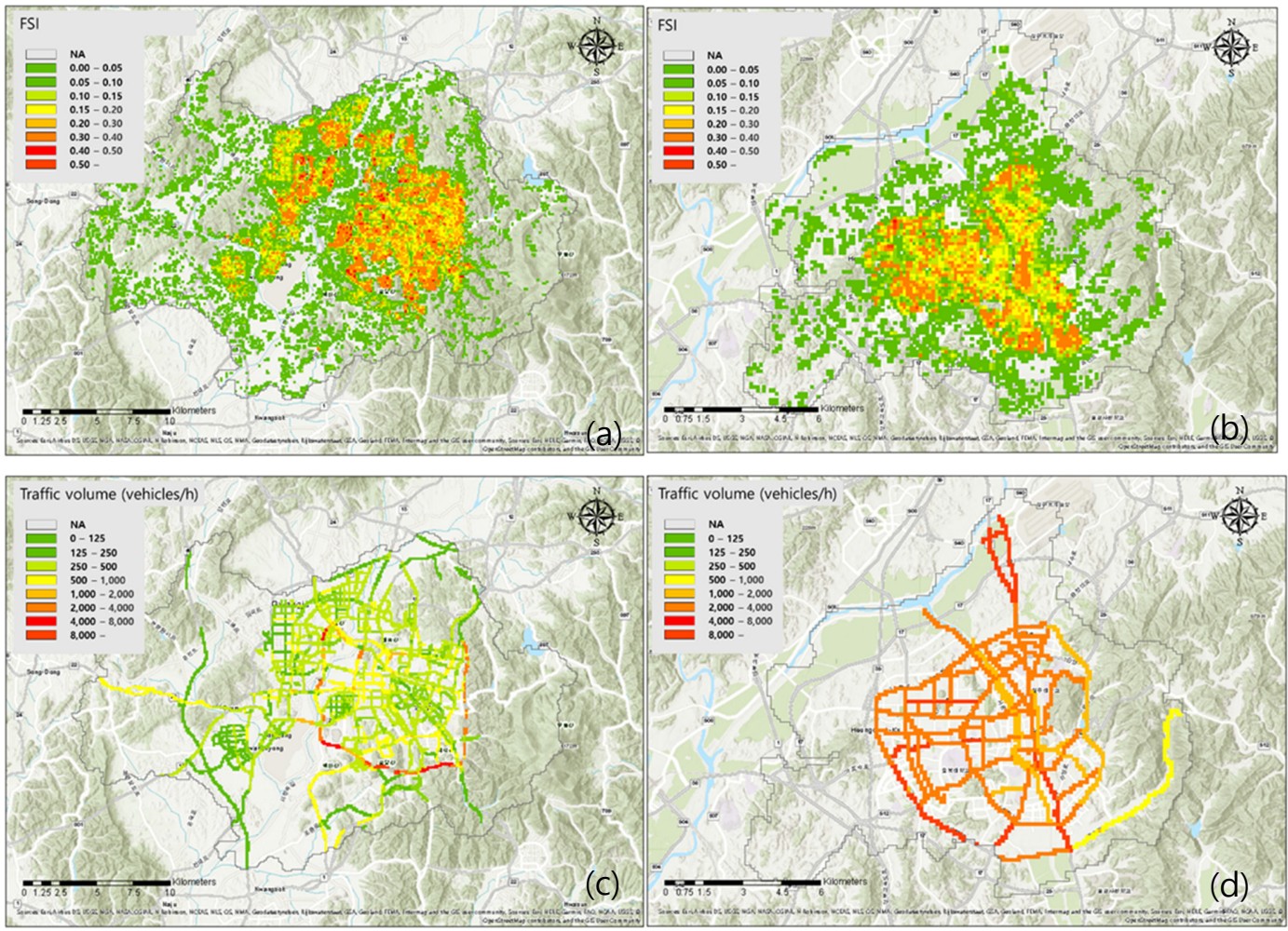

**Figure 4.** Grid maps of: (**a**) *FSI* (floor space index) in Gwangju; (**b**) *FSI* (floor space index) in Cheongju; (**c**) *Q* (traffic volume) in Gwangju; (**d**) *Q* (traffic volume) in Cheongju. [44,45].

## 3. Results and Discussion

In this section, the relationship between the road-traffic noise level and urban form indicators was analyzed statistically by the ANN and OLS methods. The developed models were also applied to another city, Cheongju, to generate statistical noise maps which were compared to a normal noise map of Cheongju.

### 3.1. ANN and OLS Model Development

The data set for Gwangju was divided by 2 to 1 into the training and test sets, respectively. The training data set was used to analyze the relationships between road-traffic noise and the urban form indicators and to create an ANN model to predict the road-traffic noise levels with the urban form indicators.

Among the urban form indicators defined in Section 2.2, independent explanatory variables were selected to be used to develop the OLS and ANN models. The fraction of green area, $L_G$, was excluded because the degree of freedom of land-use-related urban form indicators was 3, as defined in Equation (12). The urban form indicators $D_t$ and $L_I$, which showed multi-collinearity, were excluded. Table 1 shows the result for the multi-collinearity test after the urban form indicators $D_t$ and $L_I$ were excluded, and the remaining ten indicators were used as the explanatory variables for the OLS and ANN models.

**Table 1.** Multi-collinearity test results for urban form indicators.

| Urban FormIndicators | $P$ | $GSI$ | $FSI$ | $Q$ | $PH$ | $V$ | $D_t$ | $R_a$ | $W_a$ | $L_R$ | $L_C$ | $L_I$ | $L_G$ |
|---|---|---|---|---|---|---|---|---|---|---|---|---|---|
| VIF (Variance Inflation Factors) | 3.94 | 2.41 | 5.00 | 1.57 | 3.46 | 4.98 | - | 2.46 | 1.09 | 1.71 | 1.48 | - | - |

The results for the OLS model are presented in Appendix B. Among the ten indicators, seven statistically significant urban form indicators in the OLS model were found: $FSI$, $Q$, $PH$, $V$, $R_a$, $L_R$, and $L_C$. The urban form indicators showing statistical significance in common with Ryu et al. [25] were $Q$, $V$, $R_a$, and $L_R$. $FSI$ was a significant variable in Salomons and Berghauser Pont [23] and in this analysis. The variables related to road area, $R_a$ for this study, were significant in this and previous studies [16,19]. The population-related variables were not significant in this study or in Aguilera et al. [16].

In optimizing the process of the ANN model, the control parameters of the ANN model were adjusted until the difference between the noise levels from the normal noise map and those from the ANN model was minimized. R [50] contains various packages that can be used to optimize the ANN model, and for this study, the *nnet* package [51], which is specialized to optimize the single-hidden layer ANN model, was used. The *nnet* package can adjust various control parameters to develop the ANN model, and among them, the number of hidden nodes and the decay parameter were adjusted to examine the prediction accuracy. The accuracy was determined by calculating the root mean square error (RMSE) using the difference between the existing and predicted answers. The RMSE for the ANN model was calculated while adjusting the number of hidden nodes from 1 to 50, and the decay parameters were tested for each number of hidden nodes from $e^{-2}$ to $e^7$, as well as 0. The thirty-three hidden nodes and decay parameter, $e^{-1}$, that had the lowest RMSE were applied to the final ANN model. Figure 5 shows the RMSE test results. As an activation function, *linout* [51] was selected. The *maxit* [51] that determines the number of iterations was set to 200. Figure 6 shows the final ANN model, with 10 input nodes for urban form indicators, 33 hidden nodes, and 1 output node.

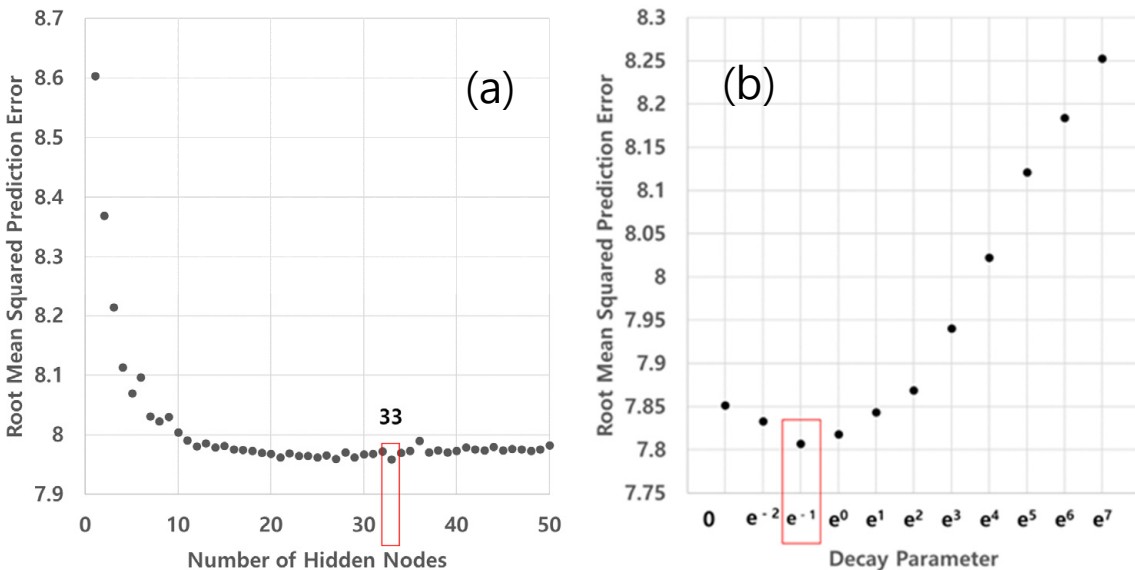

**Figure 5.** Minimal root mean square error (RMSE) investigation to optimize the artificial neural network (ANN) model parameters: (**a**) number of hidden nodes; (**b**) decay parameter (number of hidden nodes = 33).

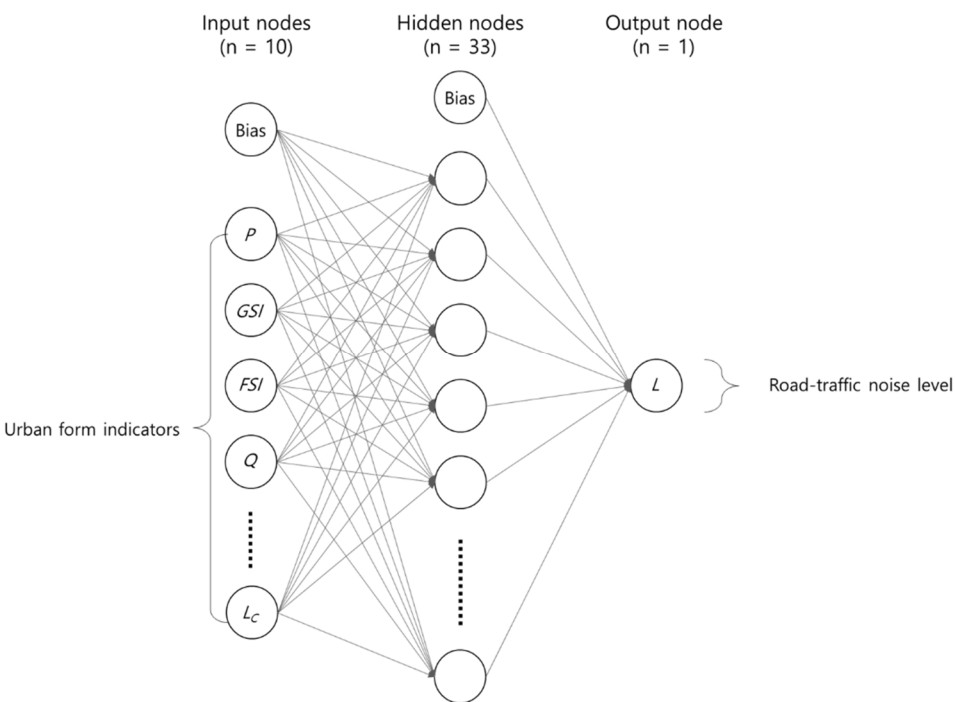

**Figure 6.** Artificial neural network model.

### 3.2. Statistical Noise Mapping by the ANN and OLS Models

The ANN and OLS models, which were developed from the dataset of Gwangju, were applied to the original city, Gwangju, to test the models. The correlation coefficient and coefficient of determination for predicted road-traffic noise levels between the normal noise map and the ANN model were 0.70 and 0.49, respectively, and for the predicted levels between the OLS model and the noise map were 0.66 and 0.44, respectively. These findings demonstrate that the statistical noise map created by using the ANN model was slightly more accurate than the OLS map. Figure 7a,b show the ANN and OLS statistical grid maps for the road-traffic noise level in Gwangju. The two statistical noise maps are comparable with the grid map of the road-traffic noise level from the normal noise map in Figure 3a. It was found that the OLS model tends to under-predict the road-traffic noise level, while the ANN model tends to over-predict it. The corresponding results of the noise-exposed population from the two models are presented in Appendix C.

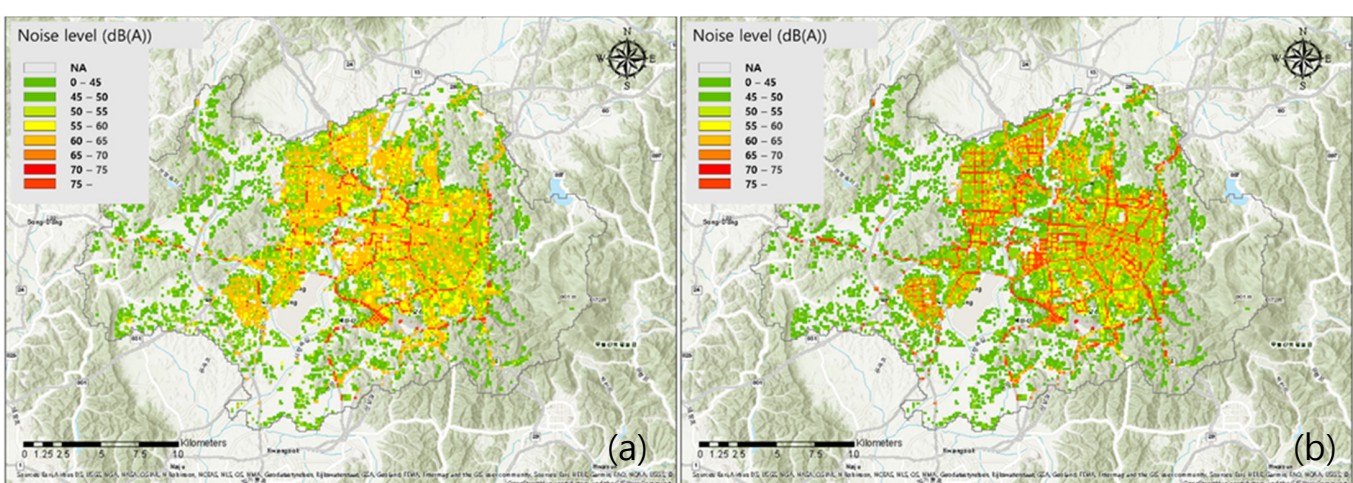

**Figure 7.** Statistical road-traffic noise maps of Gwangju: (**a**) ANN model; (**b**) ordinary least squares (OLS) model. [44,45].

The ANN and OLS models were applied to Cheongju, and the predicted road-traffic noise levels were compared to the levels predicted by a normal noise map. The correlation coefficient and coefficient of determination for predicted road-traffic noise levels in Cheongju between the ANN model and the normal noise map were 0.66 and 0.44, respectively, and the correlation coefficient and coefficient of determination between the OLS model and the normal noise map were 0.70 and 0.49, respectively. Unlike for Gwangju, the correlation coefficient and coefficient of determination by the OLS model were higher than those from the ANN model.

Figure 8a,b show the statistical grid maps for road-traffic noise levels in Cheongju using the ANN and OLS models, respectively. The two statistical noise maps were comparable with the grid map of representative values of road-traffic noise levels from the normal noise map in Figure 3b. As in Gwangju, the OLS model tended to under-predict the road-traffic noise level, while the ANN model tended to over-predict it. The corresponding results of the noise-exposed population from the two models are presented in Appendix C.

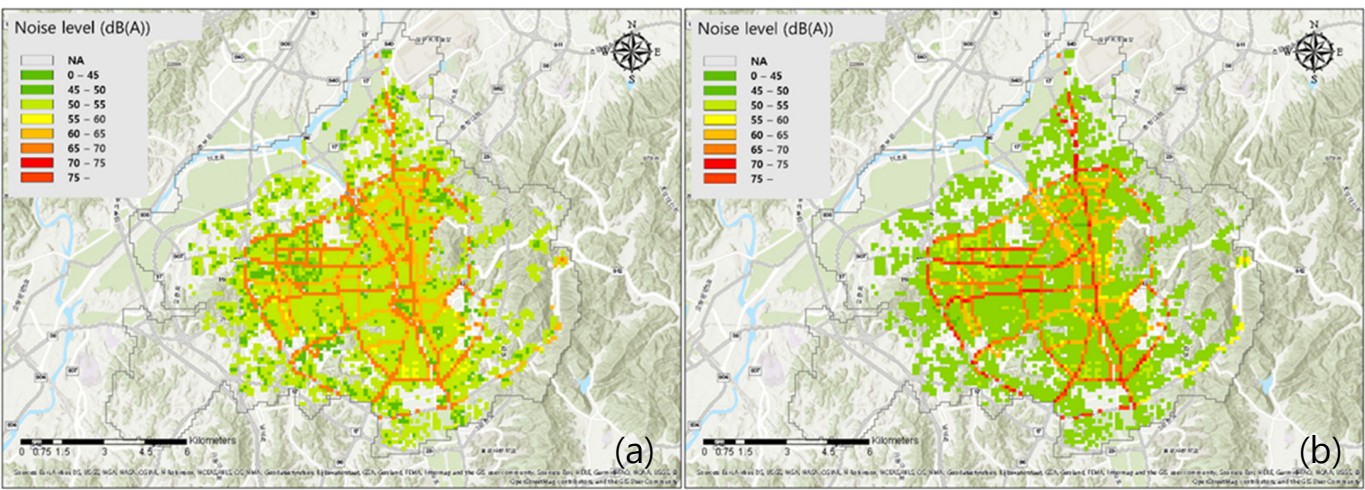

**Figure 8.** Statistical road-traffic noise map of Cheongju: (**a**) ANN model; (**b**) OLS model [44,45].

### 3.3. Importance of Urban Form Indicators in the ANN Model

The effects of urban form indicators on road-traffic noise prediction in the ANN model were tested by excluding urban form indicators one by one. Ten test ANN models were developed using only nine urban form indicators, excluding one urban form indicator, and the excluded urban form indicator was the subject of a test. RMSE differences between each test model and the complete ANN model, which are shown in Figure 6, were compared. The larger the difference in the RMSE, the more important the excluded indicator was. Figure 9 shows the RMSE differences for each urban form indicator.

This RMSE difference analysis showed that traffic volume in the traffic-related urban form indicators was the most important among all urban form indicators, as in previous research [20,25,26], where it was found to be statistically significant. In Aguilera et al. [16], however, traffic volume was not found to be an important variable, but the variables related to road area and length were significant in this analysis and Aguilera et al. [16]. While the heavy vehicle ratio in the traffic-related urban form indicators in this study was similar to the truck flow variable in Aguilera et al. [16], truck flow was the third important variable among 28 variables in their study [16], but the heavy vehicle ratio was just the fifth most important among 10 variables in this study. Road-related variables such as traffic volume, road area, or speed were commonly important or significant variables in previous research [16,17,19,25,26] and in this study.

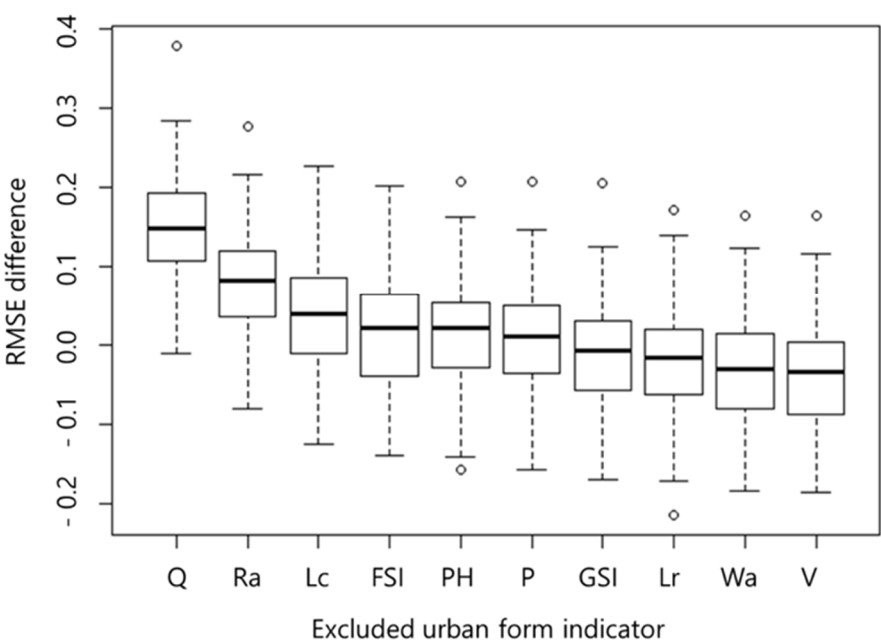

**Figure 9.** RMSE differences between the complete and test ANN models.

In Ryu et al. [25], where similar urban form indicators to this study were adopted, *GSI* (ground space index) was more significant than *FSI* (floor space index), but our analysis showed that *FSI* was more important than *GSI*. In Salomons and Berghauser Pont [23], the increases of *GSI* and *FSI* caused road-traffic noise levels to decrease. In Aguilera et al. [16], where a LUR model was developed, the average height of buildings at measurement sites was statistically significant in only one city among the three cities that were studied. The proportion of open space (porosity) and shape of the urban patch (CI) showed positive correlations with road-traffic noise, but the complexity of the perimeter of an urban area (fractal) showed a negative correlation with road-traffic noise in Silva et al. [24].

In the RMSE difference analysis, $L_C$ (fraction of commercial area) was the third most important variable among 10 urban form indicators, but in Sieber et al. [19], three land-use-related variables (household density, commercial land use, and industrial land use) were selected as predictor variables, and commercial and industrial land-use variables were not statistically significant. In Chang et al. [20], green land and agriculture-use areas were included with another four variables for the road-traffic noise level prediction. In Ryu et al. [25], where similar urban form indicators to our study were adopted, $L_I$ (fraction of industrial area) was statistically significant, but $L_C$ was not statistically significant.

## 4. Conclusions

ANN and OLS models were developed to generate statistical road-traffic noise maps based on elementary urban form indicators. The cities of Gwangju and Cheongju, which already have normal noise maps, were selected as the study and verification areas, respectively. The study area was divided into a set of square grid cells, and the representative values of the road-traffic noise level and urban form indicators were estimated for each cell. The ANN and OLS models were fitted by using the dataset of Gwangju. The ANN model was fitted by optimizing the number of hidden nodes and decay parameters.

The ANN and OLS models which were developed based on the city of Gwangju were applied to the city of Cheongju, and statistical noise maps were generated. The predicted road-traffic noise levels from the statistical models were compared to those from the normal noise maps to verify the statistical models' applicability. The predictability of the ANN and OLS models was slightly different, but it was found that the OLS model tended to under-estimate noise levels and the ANN model tended to over-estimate noise levels. The

road-related urban form indicators, such as traffic volume and road area density, were found to be important in the prediction of the road-traffic noise level by the ANN model.

The limitation of the statistical noise maps is that only one representative value of road-traffic noise level can be predicted for an area of a grid cell. The normal noise map can give the noise levels of the building facades and grids with adjustable resolutions, but the statistical noise maps, which are generated by any representative or averaged values of the urban form indicators, need a minimal area to extract them. Therefore, the statistical noise maps can only predict the noise level of a certain area, such as a grid cell.

In some previous studies which predicted environmental phenomenon such as road-traffic noise [25] and urban heat islands [40] based on urban form indicators of the grid cell system, the spatial dependency of neighboring cells was considered simultaneously with a target cell through a spatial statistical model. In this study, only urban form indicators of one target cell were considered independently to develop the ANN model, but considering the effects of neighboring cells would be useful in a future study of the convolutional neural network (CNN) model, which is used for image recognition and classification.

**Supplementary Materials:** The following are available online at https://www.mdpi.com/2071-1050/13/4/2365/s1, Figure S1. Grid maps of representative values for road-traffic noise level: (a) Gwangju; (b) Cheongju, Figure S2. Grid maps of representative values for population density: (a) Gwangju; (b) Cheongju, Figure S3. Grid maps of representative values for ground space index: (a) Gwangju; (b) Cheongju, Figure S4. Grid maps of representative values for floor space index: (a) Gwangju; (b) Cheongju, Figure S5. Grid maps of representative values for traffic volume: (a) Gwangju; (b) Cheongju, Figure S6. Grid maps of representative values for heavy vehicle ratio: (a) Gwangju; (b) Cheongju, Figure S7. Grid maps of representative values for traffic speed: (a) Gwangju; (b) Cheongju, Figure S8. Grid maps of representative values for traffic density: (a) Gwangju; (b) Cheongju, Figure S9. Grid maps of representative values for road area density: (a) Gwangju; (b) Cheongju, Figure S10. Grid maps of representative values for noise barrier density: (a) Gwangju; (b) Cheongju, Figure S11. Grid maps of representative values for fraction of residential area: (a) Gwangju; (b) Cheongju, Figure S12. Grid maps of representative values for fraction of commercial area: (a) Gwangju; (b) Cheongju, Figure S13. Grid maps of representative values for fraction of industrial area: (a) Gwangju; (b) Cheongju, Figure S14. Grid maps of representative values for fraction of green area: (a) Gwangju; (b) Cheongju, Figure S15. Statistical road-traffic noise maps of Gwangju: (a) ANN model; (b) OLS model, Figure S16. Statistical road-traffic noise map of Cheongju: (a) ANN model; (b) OLS model.

**Author Contributions:** Conceptualization, H.R. and P.K.; methodology, J.-J.J.; data curation, H.R.; writing—original draft preparation, P.K.; writing—review and editing, S.I.C. All authors have read and agreed to the published version of the manuscript.

**Funding:** This work was supported by the Basic Science Research program through the National Research Foundation of Korea (NRF), funded by the Ministry of Education (NRF-2017R1D1A1B03035253) for Phillip Kim and Hunjae Ryu. This work was also supported by the 2020 Research Fund of the University of Seoul for Seo Il Chang.

**Institutional Review Board Statement:** Not applicable.

**Informed Consent Statement:** Not applicable.

**Data Availability Statement:** Data sharing is not applicable to this article.

**Conflicts of Interest:** The authors declare no conflict of interest.

## Appendix A

**Table A1.** Summary of the Gwangju and Cheongju input data used in this research.

| City | Data Type | Parameter | Provider | Production Year | Access Type |
|---|---|---|---|---|---|
| Gwangju | Topography | Elevation | National Geographic Information Institute | 2016 | Public |
| | LiDAR | Noise barrier Point cloud | | 2007 | Proprietary |
| | Vehicle | Volume Speed Type | Gwangju Metropolitan Police Agency | 2016 | Proprietary |
| | Road | Network | Gwangju Metropolitan City Office | 2016 | Proprietary |
| | Building | Footprint Elevation Building use | Gwangju Metropolitan City Office | 2016 | Public |
| | Population | Population for "-dong" | Korean National Statistical Office | 2015 | Public |
| | Land use | Land-use classification | National Geographic Information Institute | 2017 | Public |
| Cheongju | Topography | Elevation | Chungcheongbuk-do Provincial Government | 2007 | Proprietary |
| | Vehicle | Volume Speed Type | Cheongju City Police Agency and Cheongju City Government | 2007 | Proprietary |
| | Road | Network | Chungcheongbuk-do Provincial Government | 2009 | Proprietary |
| | Building | Footprint Elevation Building use | Cheongju City Government | 2009 | Proprietary |
| | Population | Population for "-dong" | Korean National Statistical Office and Cheongju City Government | 2009 | Public and Proprietary |
| | Land use | Land-use classification | Cheongju City Government | 2007 | Proprietary |

## Appendix B

**Table A2.** Results for the OLS model.

| Variable | Estimate | Std. Error | t | Pr ($\vert$ t $\vert$) | |
|---|---|---|---|---|---|
| (Intercept) | 47.4 | 0.281 | 168.937 | $<2 \times 10^{-16}$ | *** |
| $P$ | −1.61 | 10.8 | −1.485 | 0.138 | |
| $GSI$ | 1.25 | 0.922 | 1.352 | 0.176 | |
| $FSI$ | 2.48 | 0.289 | 8.6 | $<2 \times 10^{-16}$ | *** |
| $Q$ | $2.26 \times 10^{-3}$ | $2.45 \times 10^{-4}$ | 9.229 | $<2 \times 10^{-16}$ | *** |
| $PH$ | 0.366 | $5.17 \times 10^{-2}$ | 7.093 | $1.42 \times 10^{-12}$ | *** |
| $V$ | 0.141 | $8.35 \times 10^{-3}$ | 16.932 | $<2 \times 10^{-16}$ | *** |
| $R_a$ | 28.1 | 2.12 | 13.246 | $<2 \times 10^{-16}$ | *** |
| $W_a$ | 1.4 | 11.3 | 0.124 | 0.902 | |
| $L_R$ | −1.31 | 0.314 | −4.171 | $3.06 \times 10^{-5}$ | *** |
| $L_C$ | −4.56 | 0.44 | −10.375 | $<2 \times 10^{-16}$ | *** |

Significant codes: '***' 0.001, ' ' 1. Residual standard error: 8.267 on 7982 degrees of freedom. Multiple R-squared: 0.4446, Adjusted R-squared: 0.4439. F: 639 on 10 and 7982 DF, *p*: $< 2.2 \times 10^{-16}$.

**Appendix C**

**Table A3.** Proportions of the noise-exposed population according to prediction error in Gwangju.

| Model | Prediction Error between Noise Map and Statistical Noise Map (dB(A)) | | | | | | | |
|---|---|---|---|---|---|---|---|---|
| | <−10 | −10~−5 | −5~−3 | −3~0 | 0~3 | 3~5 | 5~10 | <10 |
| ANN | 1.9% | 9.8% | 10.0% | 25.5% | 28.6% | 11.6% | 11.2% | 1.3% |
| OLS | 3.1% | 12.1% | 9.9% | 21.5% | 26.6% | 12.5% | 12.7% | 1.5% |

**Table A4.** Proportions of the noise-exposed population according to prediction error in Cheongju.

| Model | Prediction Error between Noise Map and Statistical Noise Map (dB(A)) | | | | | | | |
|---|---|---|---|---|---|---|---|---|
| | <−10 | −10~−5 | −5~−3 | −3~0 | 0~3 | 3~5 | 5~10 | <10 |
| ANN | 0.4% | 2.9% | 3.1% | 9.4% | 20.2% | 17.5% | 35.6% | 10.9% |
| OLS | 2.3% | 8.7% | 6.5% | 14.6% | 23.2% | 16.9% | 23.9% | 4.0% |

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
