# Peer review of "Statistical Road-Traffic Noise Mapping Based on Elementary Urban Forms in Two Cities of South Korea"

_sustainability, doi:10.3390/su13042365_

Round 1

Reviewer 1 Report

“Statistical Road-traffic Noise Mapping based on Elementary Urban Forms in Two Cities of South Korea” introduces some innovative ideas to the well-established concept of noise mapping. While the arguments is interesting and deserving the scientific attention, I must report that the authors have not dedicated to help the readers. The paper is really difficult to understand and follow as the English writing style is unproper and the document’s construction is not canonical. The results are that the message does not transit clear and it does not appear what are the improvements brought by it. I ask the authors to focus on the communication and give the paper a proper structure, perhaps following the journal template suggestions.

Other suggestions are:

Lines 22-39, i.e., the beginning of the introduction, have no references. They need some.

Lines 36-37 specifically need lot of references. Furthermore, please explain and summarize with few lines the works. Om the contrary, the others work reported after are too much discussed and can be more summarized.

The introduction chapter should always end with a part explaining what the paper do and why it is improving the current knowledge. Please fix it.

Chapter 2, “problem formulation” is a really unconventional name. I believe this looks more like a methodology chapter, mixed with results. However, a better explanation of what the paper will do is essential.

Conclusions and pictures’ quality must be improved

Reviewer 2 Report

The paper deal with an innovative way of producing noise maps based of unconventional methods.

While it is undeniable that the authors dedicated a lot of work to the project, I must underline that the presented document does not reflect it. In fact, the whole work is not easy followable by the readers because the structure of the paper itself does not help. Moreover, the authors forget to focus on what the paper is improving in present scientific knowledge. Details are reported to the authors.

Abstract need a sentence more in order to better introduce the topic.

The beginning of introduction should be totally rewritten as it is not coherent. I suggest something like the following: As introduced in 2002 by the European Union passed the European Noise Directive 2002/49/EC [Directive, E.U. Directive 2002/49/EC of the European parliament and the Council of 25 June 2002 relating to the assessment and management of environmental noise. Off. J. Eur. Communities L 2002, 189, 2002] and soon became the point of reference all around the world, member states are required to produce strategic noise maps for the main roads (Arana, M., San Martin, R., San Martin, M. L., & Aramendía, E. (2010). Strategic noise map of a major road carried out with two environmental prediction software packages. Environmental monitoring and assessment, 163(1-4), 503-513.), railways (Bunn, F., & Zannin, P. H. T. (2016). Assessment of railway noise in an urban setting. Applied acoustics, 104, 16-23), airports (Iglesias-Merchan, Carlos, Luis Diaz-Balteiro, and Mario Soliño. "Transportation planning and quiet natural areas preservation: Aircraft overflights noise assessment in a National Park." Transportation Research Part D: Transport and Environment 41 (2015): 1-12; Gagliardi, Paolo, et al. "ADS-B system as a useful tool for testing and redrawing noise management strategies at Pisa Airport." Acta Acustica united with Acustica 103.4 (2017): 543-551), ports (Nastasi, Marco, et al. "Parameters Affecting Noise Emitted by Ships Moving in Port Areas." Sustainability 12.20 (2020): 8742.) and urban centers (Murphy, Enda, and Eoin A. King. "An assessment of residential exposure to environmental noise at a shipping port." Environment international 63 (2014): 207-215.) every five years. Noise maps are the bases on which action plans are then designed in order to reduce noise exposure [Licitra, G., Ascari, E., & Fredianelli, L. (2017). Prioritizing process in action plans: A review of approaches. Current Pollution Reports, 3(2), 151-161.]. The scientific community was encouraged to study noise emissions, map their impact on the territory, and then mitigate the impact […add here some references about noise mitigations]. Specifically for road traffic, which is the most impacting noise source affecting human modern life style (Ruiz-Padillo, Alejandro, et al. "Selection of suitable alternatives to reduce the environmental impact of road traffic noise using a fuzzy multi-criteria decision model." Environmental Impact Assessment Review 61 (2016): 8-18;.), it is generated by many parameters (Sandberg U, Ejsmont J. Tyre/Road Noise Reference Book. Kisa, Sweden: INFORMEX; 2002), but beside the engines and the flow composition, tyre model (Licitra, G., Teti, L., Cerchiai, M., & Bianco, F. (2017). The influence of tyres on the use of the CPX method for evaluating the effectiveness of a noise mitigation action based on low-noise road surfaces. Transportation Research Part D: Transport and Environment, 55, 217-226.), pavement ageing (Licitra, Gaetano, et al. "Modelling of acoustic ageing of rubberized pavements." Applied Acoustics 146 (2019): 237-245.), pavement texture (Del Pizzo, Alessandro, et al. "Influence of texture on tyre road noise spectra in rubberized pavements." Applied Acoustics 159 (2020): 107080; Praticò, Filippo Giammaria. "On the dependence of acoustic performance on pavement characteristics." Transportation Research Part D: Transport and Environment 29 (2014): 79-87.), mixture (Praticò, Filippo G., and Fabienne Anfosso-Lédée. "Trends and issues in mitigating traffic noise through quiet pavements." Procedia-Social and Behavioral Sciences 53 (2012): 203-212; Licitra, Gaetano, et al. "Performance assessment of low-noise road surfaces in the leopoldo project: comparison and validation of different measurement methods." Coatings 5.1 (2015): 3-25) are the most important. New pavements and rubberized asphalts have been demonstrated to mitigate noise emissions, bettor before recurring to use noise barriers, which are impacting for citizens.

Please, consider the parameters affecting road traffic that I previously mention in your work.

Moreover, why the authors only studied road traffic noise and not the other sources?

Please introduce all the used acronyms the first time they are presented (e.g., GIS).

Last part of the introduction is not clear and the readers do not understand what the paper will talk about. At present, it is not clear at all what the paper is improving and how.

English should be improved. Lot of sentences or name means nothing (e.g. “Acoustically-generated road-traffic noise map”)

Pictures quality is low.

Table 1: defining a unit with something divided by itself is useless. It is just unitless.

Summary and conclusions should just be named conclusion and should be improved. They should summarize the work done, but also report the benefits of the work, its use, its negative point and the possible further improvements or future developments.

As the paper is not clear at all, and I suggest to re-edit the paper, I failed to understand if the authors performed a comparison with the so produced noise map, with a normally produced noise map. If they have not, I do not believe how it is possible to evaluate an innovative work, as declared, without any comparison with the generally used noise mapping method.

Reviewer 3 Report

The topic of the paper is interesting, certainly a relevant contribution to this special issue, and more broadly to the field of noise mapping and urban/environmental acoustics.

The level of English is generally acceptable, but further copy-editing by a native speaker would probably improve the manuscript.

The structure of Section 1 could be slightly reorganized, the parts of the literature review feel a bit disconnected at the moment. Please make clear connections between parts/paragraphs so the flow is smoother, and readability improved.

I don’t necessarily follow the definition “acoustically-generated” noise map. What is meant here? Isn’t this a normal noise map?

Section 2: when I read “Averages were calculated for the road-traffic noise levels and the urban form indicators for each grid cell” – it is not clear to me what is meant here by “average”, is this spatial average, interpolation? Please describe in more detail. In section 2.2 there is a reference to this and energy-averaged levels related to the facades are mentioned. I am not sure this is adequate in this context: is there any reference for this? Wouldn’t some kind of normalization by façade size/number be required in this case?

There is a brief mention of the effect of the grid cell size on the accuracy – but the implications of this modelling choice should be more elaborated.

I feel a proper discussion is missing in the paper – the authors should discuss more urban form indicators  and define suggestions/guidance for urban design. After reading the paper one is still left with the feeling “so what?”

Important: The quality of Figures 1, 3, 6 and 7 should be greatly enhanced as currently they are not very readable.

Round 2

Reviewer 1 Report

The paper has been improved a lot.

Reviewer 2 Report

The authors followed accurately all my instructions and suggestions. The paper is now ready for being published.

A suggestion about the pictures used in the authors' response: they are not in the paper, as it is correct, but they should be used to produce a meaningful graphical abstract. 

Reviewer 3 Report

Revisions suffice